# Synthesis and Biological Activity of *N*-acyl Anabasine and Cytisine Derivatives with Adamantane, Pyridine and 1,2-Azole Fragments

**DOI:** 10.3390/molecules27217387

**Published:** 2022-10-31

**Authors:** Gulim K. Mukusheva, Aigerym R. Zhasymbekova, Zharkyn Zh. Zhumagalieva, Roza B. Seidakhmetova, Oralgazy A. Nurkenov, Ekaterina A. Akishina, Sergey K. Petkevich, Evgenij A. Dikusar, Vladimir I. Potkin

**Affiliations:** 1Chemistry Faculty, Karaganda Buketov University, Karaganda 100024, Kazakhstan; 2Institute of Organic Synthesis and Coal Chemistry of the Republic of Kazakhstan, Karaganda 100008, Kazakhstan; 3Institute of Physical Organic Chemistry, National Academy of Sciences of Belarus, 220072 Minsk, Belarus

**Keywords:** anabasine, cytisine, amides, quaternary pyridinium salts, isoxazole, isothiazole, pyridine, adamantane, antiviral activity, antimicrobial activity, analgesic activity

## Abstract

A series of *N*-acyl derivatives of anabasine and cytisine were prepared, to discover novel, natural product-based medicinal agents. All synthesized compounds were tested for antimicrobial, antifungal, antiviral and analgesic activity. The most pronounced antibacterial activity was shown by the compounds with isoxazole fragments, while the adamantane derivatives showed the greatest antiviral effect. It was found that the majority of anabasine derivatives showed significant analgesic activity, reducing the pain response of animals to the irritating effect of acetic acid. The presence of a high level of antimicrobial and antiviral activity in newly synthesized compounds makes it possible to consider them promising for further study of their pharmacological properties.

## 1. Introduction

Currently synthetic transformations of natural compounds have been firmly established as a main route to the design of novel biologically active compounds. [1,2]. Alkaloids are one of the first plant compounds that attracted the attention of pharmacologists. Many representatives of the alkaloid class have been widely used in clinical practice for several decades, for example, the anticancer agents *Vinblastine, Kolhamin*, the antihypertensive *Vincamine, Reserpine*, the analgesic *Morphine*, the antitussive *Codeine*, and many others [3]. Alkaloids also have been intensively studied for their broad spectrum of antiviral activities against different DNA and RNA viruses [4,5].

Isoxazole, isothiazole and pyridine heterocycles are widely used structural blocks in the synthesis of new biologically active compounds, and their derivatives are widely represented among drugs and can be used for treatment of a wide variety of diseases. The inclusion of these heterocycles may contribute to the increased efficacy, decreased toxicity, and improved pharmacokinetics profiles [6,7,8,9,10]. High lipophilicity, along with the bulk structure of the adamantane radical, can significantly promote and modify their pharmacological action, due to the creation of favorable conditions for their transport through biological membranes [11].

Extending our previous studies on the synthesis of novel derivatives based on quinine alkaloid [12], we used the classical acylation reaction of anabasine and cytisine to discover new promising biologically active substances with a different spectrum of action.

Since the early pharmacological studies on cytisine and anabasine it has appeared evident that its activity is very similar to that of nicotine, suggesting that the most relevant targets of the drug are cholinergic nicotinic acetylcholine receptors (AChRs). Therefore they have been used for a long time for the treatment of tobacco addiction, and have also become popular initial matrices for the synthesis of substances with potential neurotropic properties [13,14,15,16]. However, modifying the structure of these alkaloids may reinforce some of the non-nicotinic affinities of the original molecule, and give rise to molecules with previously unknown activities, for example anticancer, antifungal, antimicrobial, antiviral activity, etc. [14,17,18,19]. These properties of anabasine and cytisine derivatives are poorly studied, and much less is known about their mechanisms of action outside the nervous system. It should be noted that alkaloids at the same time have a side toxic effect [20,21], and the substitution of hydrogen at the nitrogen of anabasine or cytisine also makes it possible to solve the toxicity problem [16,17,22,23,24,25].

Therefore, the combination of fragments of alkaloids, pyridine and 1,2-azoles in one molecule can add new useful properties to their conjugates.

## 2. Results and Discussion

### 2.1. Chemistry

In the framework of this work, new acyl derivatives of anabasine and cytisine by reaction with 1,2-azole-3-, pyridine-3-, pyridine-4- and adamantane-1-carbonyl chlorides were synthesized (Figure 1). The reaction proceeded in dichloromethane at room temperature in the presence of triethylamine, with satisfactory yields (52–87%). The lowest yield was observed for pyridine derivatives of anabasine, which is probably due to the partial solubility of the product in water.

From an analysis of ^1^H and ^13^C NMR spectra of **1a**–**d**, **2a**–**d**, **3a**–**c** we can assume the presence of two rotational isomers, cytisine and anabasine, amides with fragments of 1,2-azoles and pyridine (Figure 1) caused by inhibition of internal rotation around the C(O)-N bond. Since the barriers of these rotations are not large (Table 1), this lead to registration of spectra from both conformers and to broadening of the spectrum lines. According to ^1^H NMR spectra, the ratio of conformers is 2:3.

For alkaloid derivatives with an adamantane fragment this phenomenon is not observed in the NMR spectra, since the adamantane fragment is symmetrical, relative to the N-C(O) bond.

Based on the synthesized derivatives **1a**–**f**, quaternary pyridinium salts (iodomethylates) were obtained. Quaternization led to the formation of monoidomethylates **1a**–**c** in a 95–99% yield and diiodomethylates **2d**,**e** in a 91–95% yield. The quaternization reaction proceeded completely with a 3-fold excess of the alkylating agent, and the resulting salts precipitated out of the solution.

Quaternization of alkaloids amides makes it possible to increase the water solubility of compounds, which is important for choosing the most rational ways of introducing drugs into the body. Pyridinium salts are also known to inhibit the growth of various microorganisms such as bacteria, viruses, and fungi [26].

The obtained compounds were identified on the basis of IR, UV, mass and NMR spectra (^1^H and ^13^C), as well as elemental analysis.

In IR spectra of **1a**–**f**, **2a**–**e** characteristic bands for C–H bond vibrations of saturated fragments at 2846–2981 cm^–1^ and aromatic ones at 3058–2995 cm^–1^ are observed. Stretching vibrations of the carbonyl group appear in the spectra of all compounds as an intense band at 1613–1640 cm^–1^. IR spectra of **3a**–**c**, **f** have absorption bands for cytisine C=O bond at 1656–1657 cm^−1^. 

In the ^1^H spectra of **1a**–**f**, **2a**–**e** the signals of α-, β-, γ-protons of the anabasine pyridine ring at 8.54–8.66, 8.46–8.52, 7.22–7.49 and 7.43–7.77 ppm signals are identified, and the proton signals of the piperidine ring are located in the region of 1.26–6.20 ppm. In the downfield part of the proton spectrum of cytisine derivatives **3a**–**c**,**f** three groups of pyridine ring signals are observed (5.81–6.20, 6.20–6.46 and 7.15–7.35 ppm). The single proton of the isoxazole ring of anabasine derivatives **1b**,**c**, **2b**,**c** and the cytisine derivatives **3b**,**c** appears as a singlet in the region of 6.78–6.86 and 6.27–6.65, respectively. In the ^1^H spectra of monoidomethylates **2a**–**c** an intense signal of the methyl group appears in the region of 4.38–4.65, while the spectra of diidomethylates **2d**,**e** show two signals in the region of 4.30–4.46 ppm. In addition, four signals from the pyridine heterocycle protons appear in the ^1^H spectrum of **1d** in the region of 7.45–7.42, 7.92, 8.65, and 8.66–8.73 ppm, while in the spectrum of **1e** two signals from symmetric protons of the isonicotine fragment are in the region of 7.45–7.54 and 8.62–8.73 ppm. In the ^1^H spectra of methyl iodides **2d**,**e** all signals of pyridine heterocycles are shifted downfield by 0.4–0.53, indicating the formation of diiodomethylates. Three multiplets in the region 1.53–1.78, 1.63–1.97, and 1.82–2.14 ppm belong to the methylene protons of the adamantane fragments in the compounds **1f**, **3f**.

According to the measured optical rotation data [α]_d_^20^ the acylation reaction does not affect the configuration of the asymmetric centers of anabasine and cytisine.

All synthesized compounds were tested for antiviral, antimicrobial, antifungal and analgesic activity.

### 2.2. Evaluation of the Biological Activity

#### 2.2.1. Antimicrobial Activity

The antimicrobial activity of the samples was studied on the reference test micro-organisms recommended by the State Pharmacopoeia of the Republic of Kazakhstan (facultative anaerobic Gram-positive cocci *Staphylococcus aureus ATCC 6538*, aerobic Gram-positive spore-forming rods *Bacillus subtilis ATCC 6633*, Gram-negative facultative anaerobe rods *Escherichia coli ATCC 25922*, aerobic *Pseudomonas aeruginosa ATCC 27853* and yeast fungus *Candida albicans ATCC 10231)* using the method of diffusion into agar (wells) and also the method of serial dilutions with the determination of the minimum inhibitory concentration (MIC). 

Previously, the level of antibacterial activity of the samples was assessed by the diameter of the growth inhibition zones of the test strains (mm) around the well, including the diameter of the well itself: no growth inhibition zone indicates the test culture is not sensitive to this sample concentration; if the diameter of the zones of growth-inhibition is less than 10 mm, and there is continuous growth in the cup, this is assessed as the absence of antibacterial activity: 10–15 mm—weak activity, 15–20 mm—moderately pronounced activity, more than 20 mm—pronounced (Table 2). 

It was found that the test compounds exhibit antibacterial activity against the presented opportunistic test strains, in varying degrees. Analysis of the antimicrobial activity of the test substances showed that its manifestation depends on the type of pathogenic microorganism. Test strain *Staphylococcus aureus* is the most sensitive to all the presented compounds, except anabasine methyliodide **2a**, cytisine derivatives **3a** with isothiazole fragments and **3f** with an adamantane fragment, which was not sensitive or showed weak activity in the experiment of the growth inhibition zones of the test strains. At the same time, anabasine derivatives with adamantane and isothiazole fragments **1a**, **f** exhibit pronounced activity against *Staphylococcus aureus* as well as anabasine derivatives with the isoxazole fragment **1b**,**c** and methyliodides **2b**–**d**, showing a result comparable to the drug gentamicin *(*20–24 mm inhibition zones). 

However, the best MIC result determined by the method of serial dilutions was recorded for the anabasine and cytisine derivatives with isoxazole fragments **2b**, **2c**, **3b**, **3c** against the Gram-positive test strain of *Staphylococcus aureus ATCC 6538*, while compounds **3b**,**c** showed a moderately pronounced activity in the experiment of the growth inhibition zones (15–17 mm). The antibacterial effect of the above compounds against this test strain reached 1.3–6.7 µM (Table 3), even better than that of ceftriaxone (11 µM). Compounds **1a**–**c**, **f**, **2d** also showed a low MIC value (11.5–19.1 µM).

The obtained data on antimicrobial activity against *Staphylococcus aureus* allowed us to conclude that in some cases quaternization of *N*-acyl derivatives increased antibacterial activity, compared with the initial substrates (Figure 2), except for sample **1a** and its iodomethylate **2a**, where the opposite effect was observed. While the acyl derivative **1a** showed pronounced antibacterial effect (a 21 mm inhibition zone), its iodomethylate **2a** appeared to be not sensitive to *Staphylococcus aureus* (Table 2). A similar observation was noted for quinine esters, the results of which were published earlier [12].

According to the method of diffusion into agar (wells) the results on antibacterial activity against *Bacillus subtilis* and *Escherichia coli* were quite conflicting. Most of the compounds showed weak (14 mm) or moderately pronounced (15–17 mm) activity, but only to one of the two types of bacteria. Thus, the acyl derivative of anabasine with a 5-phenylisoxazole fragment **1b** showed a moderately pronounced activity against *Escherichia coli* and was not sensitive to *Bacillus subtilis,* while for the analog with a 4-tolyl substituent at the 5-position of the isoxazole heterocycle **1c**, the opposite situation was observed (Table 2).

The most promising for these test strains of *Bacillus subtilis* and *Escherichia coli* were the anabasine and cytisine derivatives with an isoxazole fragment **2b** and **3c**, showing activity comparable to *benzylpenicillin sodium salt* (a 14–16 mm inhibition zone), and a minimum inhibitory concentration (10.5 and 3.3 µM, respectively,) which exceeded that of the reference drug (22 and 11 µM for ceftriaxone). Compounds **1f**, **2d** also did not show high levels of the MIC value (38.5 and 45.5 µM, respectively).

The Gram-negative test strain of *Pseudomonas aeruginosa* turned out to be the most resistant to the action of these compounds. None of the test compounds showed antibacterial activity against this microorganism.

Minor antifungal activity against the yeast-like fungus *Candida albicans* was observed in the substances **1c** and **3d** at concentrations of 144.0 and 90.7 µM, respectively.

Thus, in the series of new synthesized derivatives of anabasine and cytisine, compounds with antibacterial activity comparable to the activity of the drug ceftriaxone were identified. According to the results of both methods, the derivatives with the isoxazole fragment **2b**,**c** and **3b**,**c** turned out to be the most active against *Staphylococcus aureus*. This allowed us to consider these compounds as very promising for the search for new potential antibacterial drugs, which requires further in-depth studies. 

#### 2.2.2. Analgesic Activity In Vivo

In the course of studying the analgesic activity induced by novel compounds, the animals were observed from the moment of modeling the vinegar writhing. It was found that most of the test compounds, when administered once at a dose of 25 mg/kg 1 h before the stimulus, significantly reduced the pain response of animals to the irritating effect of acetic acid (Table 4). 

The greatest analgesic effect among the studied potential pharmaceutical substances was shown by the majority of anabasine derivatives (**1a**, **1b**, **1d**, **2a**, **2b**, **2c**, **2d**, **2e**), and caused a significant decrease in the amount of vinegar writhing in mice, by 42.4, 42.9, 49.0, 44.1, 46.6, 45.2, 44.9 and 44.2%, respectively. The analgesic activity level of these compounds did not reach the level of sodium diclofenac, but was comparable to it (53.3 %).

The cytisine derivatives **3a**, **3b**, **3f** and **3c** at doses of 25 mg/kg did not show significant analgesic activity in the ″acetic writhing″ test. Thedecrease in the amount of vinegar writhing in mice varied from 22.3 to 38.4 %, which was much lower than in the case of the reference drug (Table 4).

#### 2.2.3. Antiviral Activity

##### Cytotoxicity and Chicken-Embryo Lethality of Test Compounds

At the first stage of the study, the cytotoxic effect of the samples was assessed at various doses in various in vitro models (erythrocytes, chick embryo). The interval of the dose range was determined, first of all, by the interval of acceptable values for the number of compounds used in comparable studies for antiviral activity.

The analysis of acute cytotoxicity of alkaloid derivatives “in vitro” was carried out in the concentration range of 0.03 to 1% (from 0.03 mg to 1 mg per 100 μL), corresponding to effective doses of alkaloid compounds with antiviral properties. The cytotoxicity of substances was determined by studying the effect of various doses of compounds on cell viability, using the method of dehydrogenase activity detection (MTT test). It was found that in the tested dose range, all the studied compounds failed to reach the LD50.

The analysis of the acute toxicity of compounds on the model of 10-day-old chick embryos was carried out in the dose range of 0.003–0.4 mg/chick embryo (0.06–8 mg/kg). At the maximum dose of 0.4 mg/chick embryo, the toxicity (LD50) of the test compounds was not manifested; therefore, a further study of the presence of antiviral activity was carried out in the dose range of 0.4 mg/chick embryo or less.

Thus, in the determination of acute toxicity “in vitro” and on the model of 10-day-old chicken embryos, the studied compounds did not reveal toxic properties at the maximum of the tested doses.

##### Comparative Study of the Antiviral Activity of Alkaloid-Type Compounds 

To determine the virus-inhibiting activity, influenza virus strains A/Almaty/8/98 (H3N2) and A/Vladivostok/2/09 (H1N1), with different antigenic formulas, were taken.

The virus-inhibiting activity for determining the Chemical Therapeutic Index (CTI) or Selective Index (SI) was studied at a concentration of 0.0016% to 0.2%, which corresponded to doses of 0.003–0.4 mg per chick embryo (0.06–8 mg/kg). The selective index was calculated as the ratio of the drug dose with 50% toxicity to the dose that caused 50% viral suppression. The following drugs were used as comparators: (1) Rimantadine® (Olainfarm, Olaine, Latvia); (2) Tamiflu® (Hoffmann-La Roche, Basel, Switzerland).

It was found that all the studied compounds were to some extent capable of suppressing the influenza virus reproduction (Table 5). However, when compared with the activity of commercial preparations Tamiflu and Remantadine, among the studied samples only two with high CTI had the prospect for further research. These were the compounds **1f** and **3f**, which were superior in activity to the commercial drugs. 

##### Virucidal Activity

The study of virucidal activity is one of the main approaches to determine the efficiency of drugs with antiviral activity.

The virucidal activity of the compound is associated with a direct inactivating-effect on virions; as a result, the infectivity of the virus is partially or completely lost. Virucidal activity represents the activity which functionally inhibits (neutralizes) viral infectivity, without apparent morphological alterations of the viral particles as in the case of antibody-mediated neutralization [27].

The virucidal activity of the test substances was determined by treating the virus-containing material with alkaloid derivatives at 37 °C for 30 min, followed by titration of the infectivity of the treated material. The real virucidal effect was taken as the difference between the virus titer in the sample without exposure, and its titer afterward. If the difference in titers was 1.0–2.0 lg, then the substance was considered to have moderate activity, while >2.0 lg it was considered to have pronounced antiviral activity. Infectious virus titer was determined by the method of Reed and Muench [28].

The dose of drugs was 0.4 mg/chick embryo. Influenza virus strains A/Almaty/8/98 (H3N2) and A/Vladivostok/2/09 (H1N1) were used as model viruses. It was found that the virucidal activity of the studied preparations varied from 0.25 to 1.25 lg (Table 6).

Thus, anabasine and cytisine derivatives with adamantine fragment **1f** and **3f** showed moderate antiviral activity. These compounds were able to reduce infectivity of the influenza virus by more than 1 lg, which meant a loss of 90% of the infectivity of the virus. This indicates the prospects for studying these compounds as virucidal agents that affect extracellular virions. 

## 3. Materials and Methods

### 3.1. General Chemistry Section 

UV spectra were recorded on a Varian Cary 300 spectrophotometer using quartz cuvettes with *l* = 1 cm. The concentration of the studied compounds in methanol was 4 × 10^−5^–1 × 10^−4^ M. IR spectra were registered on a Thermo Nicolet Protege 460 Fourier transform spectrometer in KBr pellets. 

^1^H and ^13^C NMR spectra were acquired on a Bruker Avance 500 spectrometer (500 and 125 MHz, respectively) in DMSO-*d*6 and CDCl_3_. The residual solvent signals (DMSO-*d*6, δH 2.5, δC 40.1 ppm; CDCl_3_, δH 7.26, δC 77.2 ppm) were used as the internal standard. The assignment of signals in the ^13^C NMR spectra was performed using the DEPT technique.

Liquid chromatography–mass spectrometry spectra were recorded on an Agilent 1200 LC-MS system, with an Agilent 6410 Triple Quad Mass Selective Detector with electrospray ionization in the positive ion registration mode (MS2 scanning mode). An Agilent ZORBAX Eclipse XDB-C18 (4.6 × 50 mm, 1.8 μm) column was used. The mobile phase was MeCN–H_2_O + 0.05% HCO_2_H, with gradient elution from 40 to 90% MeCN in 10 min. A flow rate of 0.5 mL/min was used. 

Elemental analysis was performed on a Vario MICRO cube CHNS-analyzer. The halogen content was determined by classical microanalysis, using a modified Pregl’s method. Melting points were determined on a Kofler bench. 

The optical activity of the compounds was measured on a polarimeter MCP 100 Anton Paar.

Reagents and solvents used were of analytical grade, with the content of the main component being more than 99.5%. Triethylamine (99.5%, EKOS-1) did not require additional purification. Dichloromethane (99.8%, EKOS-1) was preliminarily kept for 1 day over CaCl_2,_ to remove 0.5% of the ethanol used for stabilizing dichloromethane. 5-Arylisoxazole-3-carboxylic and 4,5-dichloroisothiazole-3-carboxylic acids and acid chlorides were synthesized according to previously described procedures [24].

(−)-Anabasine (a colorless viscous liquid, turning yellow in air and in the light; bp 276 °C at 760 mmHg, 104–105 °C at 2 mmHg; d_20_ 1.0455, n_D_ 1.5430, [α]_D_^20^ −82°) was isolated from anabasine hydrochloride (a commercial product of Shymkentbiopharm, Shymkent, Kazakhstan) as an individual isomer.

(−)-Cytisine (a commercial product of Shymkentbiopharm, Kazakhstan) is a white crystalline substance that crystallizes from acetone in rhombic prisms, mp. 153 °C, [α]_D_^20^ = −119 ° (in aqueous solution). 

### 3.2. In Vitro Biological Assays

#### 3.2.1. Antimicrobial Activity

The antimicrobial activity of the samples was studied on the reference test micro-organisms recommended by the State Pharmacopoeia of the Republic of Kazakhstan: facultative anaerobic Gram-positive cocci *Staphylococcus aureus ATCC 6538*, aerobic Gram-positive spore-forming rods *Bacillus subtilis ATCC 6633*, Gram-negative facultative anaerobe rods *Escherichia coli ATCC 25922,* aerobic *Pseudomonas aeruginosa ATCC 27853* and yeast fungus *Candida albicans ATCC 10231,* using the method of random dilutions with the determination of the minimum inhibitory concentration (MIC) [29,30] and the agar diffusion method. The test strains of microorganisms used in the study were obtained from the American Type Culture Collection.

For the serial dilution method, suspensions of test strains at a concentration of 10^6^ CFU/mL were used A suspension of test strains of microorganisms was prepared from daily cultures grown on slant agar at a temperature of 37 °C for 24 h, for the yeast fungus *Candida albicans* at 30 °C for 48 h. The antimicrobial activity of the samples was studied at dilutions in the range of 1.56–50 μg/mL. 0.1 mL of microbial suspension at a concentration of 10^6^ CFU/mL was added to each test tube, with a working dilution of each test sample. The procedure was repeated for all test cultures. A suspension of microbes with a nutrient medium without a sample was placed in control tubes. The mixture was incubated in a thermostat for 24–48 h, depending on the class of the microorganism. Following this, visually determining the presence of turbidity in each of the tubes, we chose the one that contained a clear suspension and the lowest concentration of the antimicrobial agent. This concentration was taken as the minimum bactericidal concentration. All experiments were carried out three times.

The agar diffusion method is based on the assessment of the growth inhibition of test microorganisms by certain concentrations of the test agent.

For the study, pure cultures of test strains were taken, which were preliminarily grown in a liquid medium pH 7.3 ± 0.2, at a temperature of 30 to 37 °C, for 24–48 h on slant meat-peptone agar. A standard bacterial suspension was prepared by diluting the culture 1:1000 in a sterile 0.9% isotonic sodium chloride solution. A total of 1.0 mL of the corresponding bacterial suspension was added to cups with appropriate elective, nutrient media for the studied test strains, and inoculated according to the “solid lawn” method. After drying, wells 6.0 mm in size were formed on the agar surface, into which 20 μL of the test sample (c = 1 mg/1 mL) was added. In the control, water for injection was used, which was used to dilute samples in equivolume amounts. The cultures were incubated at 37 °C for 24 h for the bacterium, and at 30 °C for 48 h for the yeast *Candida albicans*.

The antimicrobial activity of the sample was assessed by the diameter of the growth inhibition zones of the test strains (mm) around the well, including the diameter of the well itself. No growth inhibition zone means the test culture is not sensitive to this sample concentration; the diameter of the zones of growth inhibition is less than 10 mm and there is continuous growth in the cup, was assessed as the absence of antibacterial activity; 10–15 mm—weak activity, 15–20 mm—moderately pronounced activity, more than 20 mm—pronounced. Each sample was tested in three parallel experiments. Statistical processing was carried out by parametric methods with the calculation of the arithmetic mean and standard error.

The antibacterial drugs benzylpenicillin sodium salt, gentamicin, ceftriaxone and the antifungal drug nystatin were used as reference drugs.

#### 3.2.2. Analgesic Activity In Vivo

The experimental part was carried out in accordance with the “Rules of the European Convention for the Protection of Vertebrate Animals used for Experimental and Other Scientific Purposes” and in accordance with the requirements for the study of new pharmacological substances [31]. The analgesic effect of the synthesized compounds was carried out using chemical stimulus on outbred white mice, with weights in the range of 23 to 35 g. The experimental animals were kept in standard vivarium conditions, on a normal diet. Five groups containing six animals each were formed (control, reference drug “diclofenac sodium”, three novel substances).

The analgesic effect of the samples was evaluated in the chemical irritation test of the peritoneum (the “vinegar cramps” test). The abdominal constriction test was a visceral- inflammatory-pain model (acute peritonitis model). When visceral receptors were irritated with acetic acid, abdominal muscle contraction, hind limb extension and body elongation were observed [32]. A 0.75% solution of acetic acid was injected intraperitoneally in an amount of 0.1 mL per 10 g of animal weight. The potential pharmaceutically active substances were injected intragastrically at a dose of 25 mg/kg 30 min before the administration of the acetic acid. Immediately after the introduction of the stimulus, the latent time of the onset of the pain reaction “writhing” was recorded, and the writhings were counted for 30 min. The analgesic effect of the compounds was determined by the ability to reduce the number of “writhings” counted for 10, 15, 20 and 30 min, compared with the corresponding indicators in the animal control group. The model drug was the non-steroidal anti-inflammatory drug diclofenac sodium, which was tested at an effective dose of 8 mg/kg (ED_50_ = 8 mg/kg). Control animals received the equivalent volume of starchy mucus.

Analgesic activity was expressed as a percentage reduction in the number of acetic writhings in experimental rats compared to controls.

Statistical processing was carried out by parametric statistical methods, with the calculation of the arithmetic mean and standard error. Differences were considered significant at the achieved significance level *p* < 0.05.

The analgesic activity level of this compound was comparable to sodium diclofenac.

#### 3.2.3. Antiviral Activity

##### The Study of Drug Toxicity

The toxicity of the samples was studied by the treatment of 2% rooster erythrocytes, as well as on the primary culture of chick embryo fibroblasts and 10-day-old chick embryos.

The drug was diluted in a minimum volume of DMSO. A series of 2-fold dilutions was then prepared in phosphate buffer (pH 7.2).

It was shown that the maximum dose of the samples, soluble in alcohol or DMSO, was 100 mg/mL, so the doses not exceeding these values were used for further studies. Serial dilutions of the drug (10% stock solution in alcohol) were diluted in the buffer pH 7.2.

The primary determination of the toxic (hemolytic) dose of the samples was carried out using 2% of rooster erythrocytes. Therefore, the test samples were mixed in a ratio of 1:5 with a 2% solution of rooster erythrocytes. After 120 min of incubation at 37 °C, an equal volume of cold saline was added, centrifuged for 5 min at 13,000 rpm. The optical density of the supernatant liquid was measured on the M200 spectrophotometer (Tecan, Switzerland) at 412 nm. Based on the data obtained, the toxic dose of the drug (TK50) was calculated, at which 50% erythrocyte lysis occurred. Based on the TK50 value, the working concentrations of the drug were calculated.

The effect of the test sample at different doses on cell viability (primary culture of chick embryo fibroblasts, 10^4^ cells/well) was determined by the detection of dehydrogenase activity (MTT assay). The MTT assay is based on the ability of live cell dehydrogenases to reduce non-stained forms of 3-4,5-dimethylthiazol-2-yl-2,5-diphenylterarazole (MTT reagent) to blue crystalline farmazan, soluble in dimethyl sulfoxide.

An MTT solution (Calbiochem, San Diego, CA, USA) was prepared, using physiological saline at a concentration of 0.5 µg/mL. The MTT solution was added to wells with cells previously washed with the medium in a volume of 0.1 mL. After 1 h of contact of the MTT with cells, the wells were washed and filled with 0.1 mL of DMSO, after which the optical density in the wells was measured on an M200 spectrophotometer (Tecan, Zürich, Switzerland) at a wavelength of 535 nm. Based on the data obtained, the toxic dose of the drug (TK50) was calculated, at which 50% cell destruction occurred. Based on the TK50 value, the working concentrations of the drug were calculated.

The toxicity of the test samples at different doses in relation to 10-day-old chicken embryos (embryotoxicity) was determined by inoculating 0.2 mL of the test compounds into the chorioallantoic cavity of chicken embryos. The toxicity of the preparations was determined by the death of chicken embryos within 4 days after the inoculation of materials.

##### Study of the Virus-Inhibiting Activity of the Samples against Influenza Viruses (H1N1, H3N2 Strains) on a Chicken Embryo Model

The specific virus-inhibiting activity of the studied compounds was determined in accordance with the methodological recommendations of the ″Guidelines for conducting preclinical studies of drugs″. Different doses of the drug were mixed with an equal volume of 100 EID_50_/mL of virus. After 30 min of incubation at 37 °C, the mixture was inoculated into 10-day-old chicken embryos. Viruses were grown in the allantoic cavity of 10-day-old chicken embryos for 24–48 h (depending on the virus strain), at 37 °C. The presence of the virus was determined by the hemagglutination test (HA). The suppression of virus reproduction was assessed by comparing the results of HA in experimental and control samples. Physiological saline was used as a control solution, pH 7.2. In line with the results of the experiments, the average effective virus-inhibiting concentration of the test drug (EC_50_) was determined.

The chemotherapeutic index (CTI) was calculated as a criterion for the specific antiviral action of the compounds, by the ratio of the median toxic concentration of the substance (TC_50_) to the median effective virus-inhibiting dose (EC_50_).

##### Evaluation of the Ability to Suppress the Infectivity of the Influenza Virus (on a Model of 2 Strains)

Determination of virucidal activity. The samples in different doses were mixed with an equal volume of influenza virus with the infectivity titer of at least 10^−7^ EID_50_/0.2 mL. The mixture was incubated for 30 min, at 37 °C. A number of consecutive 10-fold dilutions were prepared,starting from the maximum dilution; 0.2 mL were infected in 10-day-old chick embryos (at least 4 chick embryos). After 24–48 h, the presence of the virus was determined by hemagglutination in the allantoic fluid of the embryo.. Cumulative data (accumulative) were determined. When calculating the cumulative data of the number of healthy embryos, the higher figure was added to the lower one, starting from the smallest. It was believed that if the embryo did not show the presence of the virus with a higher dose, then it would not show with a lower dose, either. When calculating the cumulative data on the number of dead embryos, the lower figure was added to the higher one. It was believed that if the embryo was infected with a lower dose, then it would become infected with a higher dose. 

The effectiveness percentage of infection was determined as follows: cumulative number of infected × 100/(cumulative number of infected + cumulative number of healthy). 

The coefficient of proportionality (CP) was determined:(1)$$CP=\frac{\left(\;infection\;percentage\;with\;HCD-50\right)\;\;\;\;\;\;\;\;\;\;\;\;\;\;\;\;\;\;\;\;\;\;\;\;\;\;\;\;\;\;\;\;\;\;\;\;\;\;\;\;\;\;\;\;\;\;\;\;\;\;\;\;\;\;\;\;\;\left(54-50\right)}{\left(\;infection\;percentage\;with\;HCD-\;infection\;percentage\;with\;LCD\right)\;\;\;\;\left(54-29\right)}=0.16$$

10^−8^—the highest critical dose (HCD), 54%

10^−9^—the lowest critical dose (LCD), 29%

T = HCD—CP = 10^−8–0,16^= 10^−8,16^ = EID_50_/0.2 mL.

8—shows how many signs will be in the final result (+1).

16—mantissa. Using the mantissa, according to the table of antilogarithms we find the number (1445) 1:144,500,000 EID_50_/0.2 mL or 1 EID_50_ = 1:144,500,000 in 0.2 mL.

If the infectivity titer is suppressed by more than 1 lg, then the substance is considered capable of suppressing 99% of the virus in the sample.

### 3.3. Quantum Chemical Calculations 

In this work, ab initio quantum-chemical calculations were applied. Calculations were carried out using the DFT method, using the B3LYP1/MIDI theory level with the GAMESS software package [33] and the MIDI basis set [34].

### 3.4. Experimental Section

#### 3.4.1. General Procedure for the Synthesis of anabasine Derivatives **1a**–**c**,**f**

Anabasine (1.6 g, 10 mmol) was dissolved in 100 mL of dry dichloromethane.Following this, 1.2 g (12 mmol) of triethylamine and 11 mmol of 1,2-azole-3- (**1a**–**c**) or adamantane- (**1f**) carbonyl chlorides were successively added to the resulting solution, under stirring. The mixture was stirred for 1 h and left for 15 h at 20–23 °C. The mixture was washed with water (2 × 200 mL, 1 h stirring) and 5% sodium bicarbonate solution (2 × 200 mL, 1 h stirring). The organic layer was separated and dried over anhydrous Na_2_SO_4_. The solvent was removed, and the residue was crystallized from a mixture of ether and hexane (1:3).

*(S)-(4,5-dichloroisothiazol-3-yl)(2-(pyridin-3-yl)piperidin-1-yl)methanone* (**1a**): white solid; yield 82%; mp 154–155 °C; [α]_D_^20^ –183.5°; UV (MeOH c = 6 × 10^−5^ M) λ_max_ (log ε) 257 (3.95), 263 (4.04), 269 (4.00); IR (KBr) ν 3262, 3058, 3041 (C=C-H); 2981, 2950, 2935, 2872 (C–H_aliph_), 1637 (C=O), 1587, 1572, 1500 (C=C_arom_); 1448, 1440, 1416, 1346, 1319, 1253, 1245, 1192, 1163, 1129, 1014, 962, 825, 808, 712, 691, 641, 620, 571, 548, 493 cm^−1^; ^1^H NMR (DMSO–*d*6, 500 MHz) δ 1.35–1.72 (4H, m, 2CH_2_), 1.81–1.95, 2.40–2.56 (2H, m, CH_2_); 2.61–2.73, 2.89–3.01 (1H, m, NCH_2_); 3.33–3.44, 4.39–4.49 (1H, m, NCH_2_); 4.96–5.04, 5.86–5.93 (1H, m, CH); 7.38–7.49 (1H_Py_, m); 7.67–7.77 (1H_Py_, m); 8.46–8.61 (2H_Py_, m); ^13^C NMR (DMSO-*d*6, 125 MHz) δ 19.58, 19.65 (CH_2_); 25.53, 26.08 (CH_2_); 27.45, 28.77 (CH_2_); 38.65, 43.72 (NCH_2_); 50.12, 55.39 (CH); 124.18, 124.33 (1CH_Py_); 134.92, 135.04 (1CH_Py_); 148.57, 148.70 (2CH_Py_); (121.65, 121.78); (134.24, 134.35); (149.59, 149.73); 160.68; (162.07, 162.17) (5C_quater_); MS *m*/*z* (*I*_rel_, %) 342.00 [M+H]^+^ (100); Anal. calcd. for C_14_H_13_Cl_2_N_3_OS (342.24): C, 49.13; H, 3.83; Cl, 20.72; N, 12.28; S, 9.37%; Found: C, 49.42; H, 4.01; Cl, 20.41; N, 12.00; S, 9.03%.

*(S)-(5-phenylisoxazol-3-yl)(2-(pyridin-3-yl)piperidin-1-yl)methanone* (**1b**): oil; yield 87%; [α]_D_^20^ –140.8°; UV (MeOH c = 1 × 10^−4^ M) λ_max_ (log ε) 264 (4.36); IR (KBr) ν 3121 (CH_isox_), 3037 (C=C–H); 2942, 2865 (C–H_aliph_); 1640 (C=O), 1590, 1573, 1479 (C=C_arom_); 1447, 1420, 1395, 1261, 1237, 1148, 1127, 1021, 981, 949, 851, 767, 708, 691 cm^−1^; ^1^H NMR (CDCl_3_, 500 MHz) 1.35–1.81 (4H, m, 2CH_2_), 2.00–2.11, 2.44–2.52 (2H, m, CH_2_); 2.66–2.76, 2.99–3.12 (1H, m, NCH_2_); 4.29–4.42, 4.61–4.73 (1H, m, NCH_2_); 5.88–5.98, 6.06–6.18 (1H, m, CH); 6.84, 6.86 (1H_isox_, s); 7.29–7.34 (1H_Py_, m); 7.40–7.51 (3H_Ar_, m); 7.61–7.68 (1H_Py_, m); 7.70–7.84 (2H_Ar_, m); 8.52 (1H_Py_, d, *J* = 4.5 Hz); 8.55–8.66 (1H_Py_, m); ^13^C NMR (CDCl_3_, 125 MHz) δ 19.62 (CH_2_); 25.55, 26.41 (CH_2_); 27.07, 28.39 (CH_2_); 39.06, 43.89 (NCH_2_); 50.36, 55.37 (CH); 100.73, 100.77 (CH_isox_); 123.77 (1CH_Py_); 126.05 (2CH_Ar_); 129.24 (1CH_Ar_); 130.81 (2CH_Ar_); 134.84, 134.93 (1CH_Py_); 148.39, 148.43 (1CH_Py_); 148.53, 148.60 (1CH_Py_); (126.73, 126.82); (134.15, 134.28); (159.21, 159.29); (160.80, 160.98); 170.63 (5C_quater_); MS *m*/*z* (I_rel_, %) 334.20 [M+H]^+^ (100), 356.10 [M+Na]^+^ (6.0), 667.30 [2M+H]^+^ (34.3), 689.30 [2M+Na]^+^ (34.2); Anal. calcd. for C_20_H_19_N_3_O_2_ (333.39): C, 72.05; H, 5.74; N, 12.60%; Found: C, 72.35; H, 5.81; N, 12.44%.

*(S)-(2-(pyridin-3-yl)piperidin-1-yl)(5-(p-tolyl)isoxazol-3-yl)methanone* (**1c**): oil; yield 78%; [α]_D_^20^ –146°; UV (MeOH c = 1 × 10^−4^ M) λ_max_ (log ε) 270 (4.40); IR (KBr) ν 3128 (CH_isox_), 3128, 3032 (C=C–H), 2925, 2857 (C–H_aliph_), 1635 (C=O), 1594, 1573, 1510, 1478 (C=C_arom_), 1442, 1418, 1392, 1258, 1019, 980, 820, 706, 503 cm^−1^; ^1^H NMR (CDCl_3_, 500 MHz) δ 1.52–1.82 (4H, m, 2CH_2_); 1.98–2.11, 2.43–2.51 (2H, m, CH_2_); 2.31–2.43 (3H, m, Me); 2.64–2.77, 2.97–3.13 (1H, m, NCH_2_); 4.29–4.42, 4.61–4.73 (1H, m, NCH_2_); 5.88–5.98, 6.08–6.18 (1H, m, CH); 6.78, 6.80 (1H_isox_, s); 7.19–7.28 (2H_Ar_, m); 7.28–7.35 (1H_Py_, m); 7.57–7.66 (2H_Ar_, m); 7.67–7.72 (1H_Py_, m); 8.52 (1H_Py_, d, *J* = 4.5 Hz); 8.58–8.62 (1H_Py_, m); ^13^C NMR (CDCl_3_, 125 MHz) δ 19.62 (CH_2_); 21.63 (Me); 25.55, 26.41 (CH_2_); 27.07, 28.37 (CH_2_); 39.02, 43.87 (NCH_2_); 50.32, 55.35 (CH); 100.09 (CH_isox_); 123.75 (1CH_Py_); 125.81, 125.99 (2CH_Ar_); 129.91, 130.18 (2CH_Ar_); 134.90 (1CH_Py_); 148.44, 148.52 (1CH_Py_); 148.65 (1CH_Py_); 124.13, (134.15, 134.30); 141.20, 159.15; 161.08; 170.82 (6C_quater_); MS *m*/*z* (I_rel_, %) 348.20 [M+H]^+^ (100), 695.30 [2M] (6.0), 717.30 [2M+Na]^+^ (12.1); Anal. calcd. for C_21_H_21_N_3_O_2_ (347.41): C, 72.60; H, 6.09; N, 12.10%; Found: C, 72.98; H, 6.22; N, 11.95%.

*(S)-adamantan-1-yl(2-(pyridin-3-yl)piperidin-1-yl)methanone* (**1f**): oil; yield 79%; [α]_D_^20^ –90.5°; UV (MeOH c = 1 × 10^−4^ M) λ_max_ (log ε) 255 (4.51), 263 (3.40), 269 (3.30); IR (KBr) ν 3083, 3034 (C=C–H), 2938, 2906, 2852 (C–H_aliph_), 1621 (C=O), 1573, 1478, 1453, 1401, 1266, 1243, 1157, 1102, 1005, 973, 936, 715 cm^−1^; ^1^H NMR (CDCl_3_, 500 MHz) δ 1.46–1.62 (2H, m, CH_2_), 1.64–1.78 (6H, m, CH_2adam_); 1.85–1.93 (3H, m, 3CH_adam_), 1.97–2.14 (10H, m, 3CH_2adam_+2CH_2_); 2.33–2.43 (1H, m, CH_2_); 4.27–4.38 (1H, m, CH_2_); 5.88–6.02 (1H, m, CH); 7.22–7.29 (1H_Py_, m); 7.43–7.51 (1H_Py_, m); 8.41–8.52 (2H_Py_, m); ^13^C NMR (CDCl_3_, 125 MHz) δ 19.70 (CH_2_); 26.39, 27.27 (CH_2_); 27.80 (CH); 28.66 (3CH_adam_); 36.43 (CH_2_); 36,77 (3CH_2adam_); 38.40 (CH_2_); 39.30 (3CH_2adam_); 123.64 (1CH_Py_); 134.90 (1CH_Py_); 147.94 (1CH_Py_); 148.73 (1CH_Py_); 42.23, 135.24, 176.73 (3C_quater_); MS *m*/*z* (I_rel_, %) 325.30 [M+H]^+^ (100), 671.40 [2M+Na]^+^ (10.0); Anal. calcd. for C_21_H_28_N_2_O (324.47): C, 77.74; H, 8.70; N, 8.63%; Found: C, 77.98; H, 8.76; N, 8.52%.

#### 3.4.2. General Procedure for the Synthesis of Compounds **1d**,**e**

Anabasine (1.6 g, 10 mmol) was dissolved in 100 mL of dry dichloromethane. Following this, 1.2 g (12 mmol) of triethylamine and 2.5 g (25 mmol) of Et_3_N and 1.8 g (11 mmol) of hydrochloride, of nicotinic, or of isonicotinic carbonyl chlorides, were successively added to the resulting solution, under stirring. The mixture was stirred for 1 h, and left for 15 h at 20–23 °C. The mixture was washed with 5% sodium bicarbonate solution (1 × 100 mL, 0.5 h stirring). The organic layer was separated and dried over anhydrous Na_2_SO_4_. The solvent was removed, and the residue was crystallized from a mixture of ether and hexane (1:3).

*(S)-pyridin-3-yl(2-(pyridin-3-yl)piperidin-1-yl)methanone* (**1d**): white solid; yield 53%; mp 93–94 °C, mp 248°C [35]; [α]_D_^20^ –577.5°; UV (MeOH c = 7 × 10^−5^ M) λ_max_ (log ε) 248 (3.77), 257 (3.85), 262 (3.85), 270 (3.77); IR (KBr) ν 3221, 3090, 3020 (C=C-H), 2919, 2850 (C-H_aliph_), 1613 (C=O), 1588, 1570 (C=C_arom_), 1439, 1413, 1324, 1274, 1111, 1025, 998, 828, 812, 737, 712, 627 cm^−1^; ^1^H NMR (DMSO–*d*6, 500 MHz) δ 1.26–1.42 (1H, m, CH_2_), 1.48–1.58 (2H, m, CH_2_), 1.60–1.68 (1H, m, CH_2_), 1.90–1.99 (1H, m, CH_2_), 2.40–2.48 (1H, m, CH_2_), 2.74–3.20 (1H, m, CH_2_), 3.40–3.58 (1H, m, CH_2_), 3.58–5.30, 5.30–6.20 (1H, m, CH); 7.42 (1H_Py_, dd, *J* = 8.0, 4.8 Hz), 7.45–7.52 (1H_Py_, m), 7.75 (1H_Py_, d, *J* = 8.0 Hz), 7.92 (1H_Py_, d, *J* = 5.8 Hz), 8.48–8.52 (1H_Py_, m), 8.54–8.60 (1H_Py_, m), 8.65 (1H_Py_, d, *J* = 8.0 Hz), 8.66–8.73 (1H_Py_, m); MS *m*/*z* (I_rel_, %) 268.20 [M+H]^+^ (100); Anal. calcd. for C_16_H_17_N_3_O (267.33): C, 71.89; H, 6.41; N, 15.72%; Found: C, 72.18; H, 6.53; N, 15.38%. 

*(S)-(2-(pyridin-3-yl)piperidin-1-yl)(pyridin-4-yl)methanone* (**1e**): white solid; yield 52%; mp 96–97 °C, mp 100–101 °C [36]; [α]_D_^20^ –133°; UV (MeOH c = 7 × 10^−5^ M) λ_max_ (log ε) 256 (3.78), 262 (3.81), 269 (3.70); IR (KBr) ν 3043, 3020 (C=C–H), 2940, 2855 (C–H_aliph_), 1623 (C=O), 1594, 1570, 1546, 1460, 1434, 1410, 1407, 1323, 1272, 1024, 1000, 834, 710, 627, 595 cm^−1^; ^1^H NMR (DMSO–*d*6, 500 MHz) δ 1.28–1.43 (1H, m, CH_2_), 1.43–1.60 (2H, m, CH_2_), 1.60–1.71 (1H, m, CH_2_), 1.89–1.99 (1H, m, CH_2_), 2.34–2.49 (1H, m, CH_2_), 2.60–3.10 (1H, m, CH_2_), 3.20–3.45 (1H, m, CH_2_), 4.40–5.00, 5.80–6.00 (1H, m, CH); 7.43 (1H_Py_, dd, *J* = 8.0, 4.7 Hz), 7.45–7.54 (2H_Py_, m), 7.71–7.77 (1H_Py_, m), 8.51 (1H_Py_, d, *J* = 4.4 Hz), 8.53–8.59 (1H_Py_, m), 8.62–8.73 (2H_Py_, m); MS *m*/*z* (I_rel_, %) 268.20 [M+H]^+^ (100); Anal. calcd. for C_16_H_17_N_3_O (267.33): C, 71.89; H, 6.41; N, 15.72%; Found: C, 72.15; H, 6.51; N, 15.34%.

#### 3.4.3. General Procedure for the Synthesis of Methyliodides **2a**–**e**

The mixture of 6 mmol of amides **1a**–**e** in 30 mL of dry dichloromethane and 18 mmol (1 mL) of dry methyl iodide was kept for 14 days in the dark. Following this, the precipitated product was filtered off, washed with 2 × 5 mL of methylene chloride, and dried in a vacuum.

*(S)-3-(1-(4,5-Dichloroisothiazole-3-carbonyl)piperidin-2-yl)-1-methylpyridin-1-ium iodide* (**2a**): orange solid; yield 99%; mp 161–162 °C; [α]_D_^20^ –118.6°; UV (MeOH c = 7 × 10^−5^ M) λ_max_ (log ε) 218 (4.40), 266 (4.08); IR (KBr) ν 3036 (C=C–H), 2933, 2870 (C–H_aliph_), 1631 (C=O), 1590, 1504, 1464, 1446, 1355, 1328, 1289, 1250, 1171, 1010, 964, 899, 828, 738, 670 cm^−1^; ^1^H NMR (DMSO–*d*6, 500 MHz) δ 1.34–1.75 (4H, m, 2CH_2_), 1.82–1.94, 1.95–2.06 (1H, m, CH_2_); 2.43–2.56 (1H, m, CH_2_); 2.71–2.81, 3.04–3.14 (1H, m, NCH_2_); 3.46–3.55, 4.44–4.52 (1H, m, NCH_2_); 4.38–4.43 (3H, m, MeN), 5.20–5.25, 5.93–5.99 (1H, m, CH); 8.13–8.18, 8.19–8.25 (1H_Py_, m); 8.36–8.42, 8.43–8.49 (1H_Py_, m); 8.92–9.01 (2H_Py_, m); ^13^C NMR (DMSO-*d*6, 125 MHz) δ 19.25, 19.45 (CH_2_); 25.34, 25.62 (CH_2_); 27.42, 29.03 (CH_2_); 38.86, 43.85 (NCH_2_); 48.83, 48.91 (NMe); 50.38, 55.27 (CH); 128.11, 128.40 (1CH_Py_); 143.42, 143.83 (1CH_Py_); 144.54, 144.84 (2CH_Py_); (122.05, 122.15); (134.24, 134.35); 149.82; (159.93, 160.03); (161.98, 162.49) (5C_quater_); MS *m*/*z* (*I*_rel_, %) 357.10 [M–I]^+^ (17.1); Anal. calcd. for C_15_H_16_Cl_2_IN_3_OS (484.18): C, 37.21; H, 3.33; Cl, 14.64; I, 26.21; N, 8.68; S,8.97%; Found: C, 37.55; H, 3.61; Cl+I, 40.43; N, 8.39; S, 8.67%.

*(S)-1-Methyl-3-(1-(5-phenylisoxazole-3-carbonyl)piperidin-2-yl)pyridin-1-ium iodide* (**2b**): orange solid; yield 95%; mp 64–65 °C; [α]_D_^20^ –89.5°; UV (MeOH c = 7 × 10^−5^ M) λ_max_ (log ε) 222 (4.45), 267 (4.40); IR (KBr) ν 3030 (C=C–H), 2926, 2855 (C–H_aliph_), 1633 (C=O), 1589, 1571, 1500, 1472, 1445, 1391, 1254, 1129, 982, 767, 687, 672 cm^−1^; ^1^H NMR (CDCl_3_, 500 MHz) 1.49–1.75 (4H, m, 2CH_2_), 1.77–1.83, 1.93–2.11 (2H, m, CH_2_); 2.52–2.72 (1H, m, NCH_2_); 3.15–3.25, 4.19–4.24 (1H, m, NCH_2_); 4.65 (3H, s, NMe); 5.84–6.01 (1H, m, CH); 6.84, 6.94 (1H_isox_, s); 7.33–7.41 (3H_Ar_, m); 7.62–7.76 (2H_Ar_, m); 8.07 (1H_Py_, t, *J =* 6.9 Hz); 8.27–8.46 (1H_Py_, m); 9.03–9.20 (2H_Py_, m); ^13^C NMR (CDCl_3_, 125 MHz) δ 19.63 (CH_2_); 25.07, 25.40 (CH_2_); 27.39, 28.32 (CH_2_); 39.43, 44.64 (NCH_2_); 50.25 (NMe); 51.10 (CH); 100.99 (CH_isox_); 126.16 (2CH_Ar_); 128.54 (1CH_Ar_); 129.31 (2CH_Ar_); 130.96 (1CH_Py_); 144.01 (1CH_Py_); 144.27 (1CH_Py_); 144.37 (1CH_Py_); 126.64; 141.57; 158.81; 161.92; 170.85 (5C_quater_); MS m/z (I_rel_, %) 348.20 [*M-I*]^+^ (100); Anal. calcd. for C_21_H_22_IN_3_O_2_ (475.33): C, 53.06; H, 4.67; I, 26.70; N, 8.84%; Found: C, 53.44; H, 4.81; I, 26.55; N, 8.74%.

*(S)-1-Methyl-3-(1-(5-(p-tolyl)isoxazole-3-carbonyl)piperidin-2-yl)pyridin-1-ium iodide* (**2c**): orange solid; yield 99%; mp 101–102 °C; [α]_D_^20^ –97°; UV (MeOH c = 6 × 10^−5^ M) λ_max_ (log ε) 219 (4.46), 273 (4.40); IR (KBr) ν 3031 (C=C–H), 2931, 2856 (C–H_aliph_), 1636 (C=O), 1595, 1506, 1486, 1453, 1439, 1392, 1253, 1218, 1156, 1023, 983, 823, 807, 673, 667 cm^−1^; ^1^H NMR (CDCl_3_, 500 MHz) 1.33–1.77 (4H, m, 2CH_2_); 1.92–2.07, 2.50–2.55 (2H, m, CH_2_); 2.33–2.40 (3H, m, Me); 2.68–2.79, 3.07–3.19 (1H, m, NCH_2_); 4.03–4.12, 4.50–4.59 (1H, m, NCH_2_); 4.42 (3H, s, NMe), 5.63–5.70, 5.94–5.98 (1H, m, CH); 7.28, 7.33 (1H_isox_, s); 7.35–7.41 (2H_Ar_, m); 7.75–7.86 (2H_Ar_, m); 8.13–8.21 (1H_Py_, m); 8.47–8.55 (1H_Py_, m); 8.96 (1H_Py_, d, *J =* 6.0 Hz); 8.98–9.03 (1H_Py_, m); ^13^C NMR (CDCl_3_, 125 MHz) δ 19.35 (CH_2_); 21.62 (Me); 25.26, 25.63 (CH_2_); 27.61, 28.70 (CH_2_); 38.99, 44.10 (NCH_2_); 48.83 (NMe), 50.77, 55.62 (CH); 101.10 (CH_isox_), 126.31 (2CH_Ar_), 128.26 (1CH_Py_), 130.45 (1CH_Py_), 130.49 (2CH_Ar_), 148.44, 148.52 (1CH_Py_); 148.65 (1CH_Py_); 124.68, 140.37, 141.51; 159.59, (160.78, 161.44); 170.30 (6C_quater_); MS m/z (I_rel_, %) 362.20 [*M-I*]^+^ (100), 363.20 [*M+H-I*]^+^ (22.0); Anal. calcd. for C_22_H_24_IN_3_O_2_ (489.35): C, 54.00; H, 4.94; I, 25.93; N, 8.59%; Found: C, 54.41; H, 5.05; I, 25.74; N, 8.25%.

*(S)-1-methyl-3-(1-(1-methylpyridin-1-ium-3-carbonyl)piperidin-2-yl)pyridin-1-ium diiodide* (**2d**): orange solid; yield 95%; mp 260–262 °C, mp 223–224°C [35]; [α]_D_^20^ –26°; UV (MeOH c = 4 × 10^−5^ M) λ_max_ (log ε) 219 (4.59), 267 (4.00); IR (KBr) ν 3026, 2995 (C=C–H), 2930, 2972 (C–H_aliph_); 1642, 1626 (C=O); 1589, 1509, 1468, 1440, 1325, 1282, 1220, 1175, 1131, 1007, 831, 680, 671 cm^−1^; ^1^H NMR (DMSO-*d6*, 500 MHz) 1.31–1.60 (2H, m, CH_2_); 1.61–1.78 (2H, m, CH_2_); 2.02–2.16 (1H, m, CH_2_); 2.34–2.45, 2.50–2.61 (1H, m, CH_2_); 2.78–2.96, 3.07–3.23 (1H, m, CH_2_); 3.49–3.67, 4.35–4.41 (1H, m, CH_2_); 4.45 (3H, s, Me); 4.46 (3H, s, Me); 5.17–5.30, 5.88–6.00 (1H, m, CH), 8.12–8.24, 8.25–8.36 (2H_Py_, m); 8.60–8.70, 8.80–8.90 (2H_Py_, m), 8.97–9.15, 9.02–9.06 (2H_Py_, m), 9.10–9.17, 9.35–9.43 (2H_Py_, m); MS m/z (I_rel_, %) 296.20 [*M-2I*]^+^ (24.3); Anal. calcd. for C_18_H_23_I_2_N_3_O (551.21): C, 39.22; H, 4.21; I, 46.05; N, 7.62%; Found: C, 39.54; H, 4.51; I, 45.89; N, 7.22%.

*(S)-1-methyl-3-(1-(1-methylpyridin-1-ium-4-carbonyl)piperidin-2-yl)pyridin-1-ium diiodide* (**2e**): orange solid; yield 91%; mp 255–257 °C; [α]_D_^20^ –73.3°; UV (MeOH c = 4 × 10^−5^ M) λ_max_ (log ε) 219 (4.56), 266 (4.08); IR (KBr) ν 3118, 3039, 3016 (C=C–H), 2930, 2866 (C–H_aliph_); 1640, 1624 (C=O); 1511, 1462, 1439, 1320, 1278, 1218, 1157, 1001, 676, 573 cm^−1^; ^1^H NMR (DMSO-*d6*, 500 MHz) 1.27–1.40, 1.40–1.54 (2H, m, CH_2_); 1.54–1.71 (2H, m, CH_2_); 1.96–2.15 (1H, m, CH_2_); 2.30–2.39, 2.45–2.55 (1H, m, CH_2_); 2.80–2.92, 3.03–3.14 (1H, m, CH_2_); 3.27–3.35, 4.45–4.53 (1H, m, CH_2_); 4.30, 4.39 (3H, s, Me); 4.42, 4.44 (3H, s, Me); 4.95–5.01, 5.87–5.95 (1H, m, CH), 8.10–8.21 (1H_Py_, m); 8.25, 8.40 (2H_Py_, d, *J* = 6.2 Hz), 8.51, 8.63 (1H_Py_, d, *J* = 8.0 Hz), 8.94–9.02 (2H_Py_, m); 9.03, 9.18 (2H_Py_, d, *J* = 6.2 Hz); MS m/z (I_rel_, %) 296.2 [*M-2I*]^+^ (9.9), 297.3 [*M-2I+H*]^+^ (8.5); Anal. calcd. for C_18_H_23_I_2_N_3_O (551.21): C, 39.22; H, 4.21; I, 46.05; N, 7.62%; Found: C, 39.71; H, 4.33; I, 45.88; N, 7.55%.

#### 3.4.4. General Procedure for the Synthesis of Cytisine Derivatives **3a**–**c**,**f**

Cytisine (1.9 g, 10 mmol) was dissolved in 100 mL of dry dichloromethane. Following this, 1.2 g (12 mmol) of triethylamine and 11 mmol of 1,2-azole-3- (**3a**–**c**) or adamantane- (**3f**) carbonyl chlorides were successively added to the resulting solution, under stirring. The mixture was stirred for 1 h, and left for 15 h at 20–23 °C. The mixture was washed with water (2 × 200 mL, 1 h stirring) and 5% sodium bicarbonate solution (2 × 200 mL, 1 h stirring). The organic layer was separated and dried over anhydrous Na_2_SO_4_. The solvent was removed, and the residue was crystallized from a mixture of ether and hexane (1:3).

*(1R,5S)-3-(4,5-Dichloroisothiazole-3-carbonyl)-1,2,3,4,5,6-hexahydro-8H-1,5-methanopyrido[1,2-a][1,5]diazocin-8-one* (**3a**): white solid; yield 81%; mp 232–233 °C; [α]_D_^20^ –146°; UV (MeOH c = 1 × 10^−4^ M) λ_max_ (log ε) 232 (4.08), 267 (3.90), 311 (3.85); IR (KBr) ν 3022, 2995 (C=C–H), 2960, 2945, 2924, 2873 (C–H_aliph_), 1656 (C=O), 1636 (C=O), 1577, 1547, 1501, 1465, 1453, 1356, 1335, 1258, 1230, 1218, 1186, 1143, 1109, 1061, 968, 798, 742, 735, 694, 644 cm^−1^; ^1^H NMR (CDCl_3_, 500 MHz) δ 1.99–2.09 (2H, m, CH_2_), 2.42–2.49, 2.59–2.67 (1H, m, CH); 2.93–3.01, 3.16–3.22 (1H, m, CH_2_); 3.05–3.11, 3.37–3.44 (1H, m, CH_2_); 3.08–3.14, 3.45–3.51 (1H, m, CH_2_); 3.57–3.64, 3.77–3.83 (1H, m, CH); 3.81–3.94 (1H, m, CH_2_); 4.11–4.23 (1H, m, CH_2_); 4.74–4.82, 4.84–4.92 (1H, m, CH_2_); 5.81, 6.11 (1H_Py_, dd, *J* = 6.8, 0.9 Hz); 6.39–6.49 (1H_Py_, m); 7.19, 7.28 (1H_Py_, dd, *J* = 9.1, 6.8 Hz); ^13^C NMR (CDCl_3_, 125 MHz) δ 26.21, 26.29 (CH_2_); 27.44, 27.68 (CH); 34.51, 34.89 (CH); 48.22, 48.68 (CH_2_); 48.83, 49.19 (CH_2_); 52.72, 53.92 (CH_2_); 105.23, 106.00 (1CH_Py_); 117.87, 118.40 (1CH_Py_); 138.65, 139.13 (1CH_Py_); (123.24, 123.31); (147.80, 147.84); (149.07, 149.72); (159.35, 159.46); (161.61, 161.74); (163.44, 163.51) (6C_quater_); MS m/z (I_rel_, %) 370.00 [M]^+^ (100), 392.00 [M+Na]^+^ (43.2), 763.00 [2M+Na]^+^ (25.1); Anal. calcd. for C_15_H_13_Cl_2_N_3_O_2_S (370.25): C, 48.66; H, 3.54; Cl, 19.15; N, 11.35; S, 8.66%; Found: C, 48.89; H, 3.66; Cl, 19.01; N, 11.13; S, 8.58%.

*(1R,5S)-3-(5-Phenylisoxazole-3-carbonyl)-1,2,3,4,5,6-hexahydro-8H-1,5-methanopyrido[1,2-a][1,5]diazocin-8-one* (**3b**): white solid; yield 77%; mp 74–76 °C; [α]_D_^20^ –360°; UV (MeOH c = 1 × 10^−4^ M) λ_max_ (log ε) 239 (4.15), 267 (4.32), 311 (3.78); IR (KBr) ν 3113 (CH_isox_), 3055 (C=C–H), 2924, 2865 (C–H_aliph_), 1656 (C=O), 1574, 1545, 1479, 1446, 1393, 1342, 1258, 1229, 1139, 1070, 983, 797, 767, 689 cm^−1^; ^1^H NMR (CDCl_3_, 500 MHz) δ 2.06–2.13 (2H, m, CH_2_), 2.49–2.55, 2.59–2.64 (1H, m, CH); 3.03–3.08, 3.16–3.22 (1H, m, CH_2_); 3.08–3.11, 3.45–3.50 (1H, m, CH_2_); 3.11–3.15, 3.52–3.57 (1H, m, CH_2_); 3.80–3.91 (1H, m, CH); 4.22–4.30 (1H, m, CH_2_); 4.53–4.60, 4.66–4.72 (1H, m, CH_2_); 4.78–4.85, 4.92–4.98 (1H, m, CH_2_); 5.90, 6.13 (1H_Py_, dd, *J* = 6.8, 0.8 Hz); 6.37, 6.65 (1H_isox_, s); 6.46 (1H_Py_, dt, *J* = 9.1, 1.5 Hz); 7.16, 7.29 (1H_Py_, dd, *J* = 9.1, 6.8 Hz); 7.42–7.49 (3H_Ar_, m); 7.66–7.71, 7.74–7.78 (2H_Ar_, m); ^13^C NMR (CDCl_3_, 125 MHz) δ 26.51, 26.60 (CH_2_); 27.88, 28.03 (CH); 34.91, 35.30 (CH); 48.49, 48.59 (CH_2_); 49.00, 49.67 (CH_2_); 53.07, 54.14 (CH_2_); 100.45, 100.78 (CH_isox_); 105.69, 106.10 (1CH_Py_); 117.80 (1CH_Py_); 125.96, 126.18 (2CH_Ar_); 129.22, 129.27 (2CH_Ar_); 130.81 (1CH_Ar_); 138.69, 139.20 (1CH_Py_); 126.75; (148.09, 148.13); (158.56, 158.65); (160.43, 160.51); (163.46, 163.52); (170.25, 170.83) (6C_quater_); MS m/z (I_rel_, %) 362.10 [M+H]^+^ (100), 384.10 [M+Na]^+^ (50.2), 723.30 [2M+H]^+^ (5.3), 745.20 [2M+Na]^+^ (32.6); Anal. calcd. for C_21_H_19_N_3_O_3_ (361.40): C, 69.79; H, 5.30; N, 11.63%; Found: C, 70.05; H, 5.47; N, 11.49%.

*(1R,5S)-3-(5-(p-tolyl)isoxazole-3-carbonyl)-1,2,3,4,5,6-hexahydro-8H-1,5-methanopyrido[1,2-a][1,5]diazocin-8-one* (**3c**): oil; yield 84%; [α]_D_^20^ –225°; UV (MeOH c = 1 × 10^−4^ M) λ_max_ (log ε) 237 (4.04, 275 (4.32), 313 (3.70); IR (KBr) ν 3125 (CH_isox_), 3090, 3055, 3030 (C=C–H), 2922, 2853 (C–H_aliph_), 1657 (C=O), 1610, 1577, 1545, 1479, 1441, 1258, 1230, 1138, 1091, 1070, 983, 795, 678 cm^−1^; ^1^H NMR (CDCl_3_, 500 MHz) δ 2.01–2.11 (2H, m, CH_2_), 2.37 (3H, s, Me), 2.46–2.53, 2.56–2.64 (1H, m, CH); 2.99–3.09 (1H, m, CH_2_); 3.09–3.13, 3.49–3.55 (1H, m, CH_2_); 3.13–3.21, 3.39–3.48 (1H, m, CH_2_); 3.76–3.90 (1H, m, CH); 4.25 (1H, m, CH_2_); 4.45–4.53, 4.59–4.68 (1H, m, CH_2_); 4.73–4.83, 4.87–4.96 (1H, m, CH_2_); 5.89, 6.12 (1H_Py_, d, *J =* 6.3 Hz); 6.27, 6.57 (1H_isox_, s); 6.44 (1H_Py_, d, *J =* 9.1 Hz); 7.15, 7.28 (1H_Py_, dd, *J =* 9.0, 6.9 Hz); 7.23 (2H_Ar_, d, *J =* 8.0 Hz); 7.55, 7.63 (2H_Ar_, d, *J =* 8.1 Гц); ^13^C NMR (CDCl_3_, 125 MHz) δ 21.51, 21.52 (Me); 26.26, 26.35 (CH_2_); 27.70, 27.84 (CH), 34.72, 35.10 (CH); 48.35, 48.38 (CH_2_); 48.87, 49.46 (CH_2_); 52.87, 53.99 (CH_2_); 99.54, 99.93 (CH_isox_); 105.64, 105.99 (1CH_Py_); 117.60 (1CH_Py_); 125.72, 125.94 (2CH_Ar_); 129.76, 129.80 (2CH_Ar_); 138.62, 139.09 (1CH_Py_); 123.88; 141.06; (148.04, 148.12); (158.32, 158.45); (160.37, 160.53); (163.32, 163.37); (170.28, 170.83) (7C_quater_); MS m/z (I_rel_, %) 376.20 [M+H]^+^ (100), 398.10 [M+Na]^+^ (30.4), 751.30 [2M+H]^+^ (40.8), 773.30 [2M+Na]^+^ (65.1); Anal. calcd. for C_22_H_21_N_3_O_3_ (375.43): C, 70.38; H, 5.64; N, 11.19%; Found: C, 70.61; H, 5.88; N, 11.01%.

*(1R,5S)-3-(Adamantane-1-carbonyl)-1,2,3,4,5,6-hexahydro-8H-1,5-methanopyrido[1,2-a][1,5]diazocin-8-one* (**3f**): white solid; yield 81%; mp 203–204 °C; [α]_D_^20^ –224°; UV (MeOH c = 1×10^−4^ M) λ_max_ (log ε) 233 (3.90), 312 (3.85); IR (KBr) ν 3052 (C=C–H), 2941, 2913, 2898, 2863, 2846 (C–H_aliph_), 1657 (C=O), 1619 (C=O), 1570, 1548, 1430, 1399, 1354, 1337, 1225, 1217, 1179, 1142, 1099, 1051, 817, 806 cm^−1^; ^1^H NMR (DMSO-*d6*, 500 MHz) δ 1.53−1.63 (6H, m, 3CH_2adam_), 1.63−1.71 (6H, m, 3CH_2aдaM_), 1.82−1.88 (3H, m, 3CH_adam_), 1.88–2.03 (2H, m, CH_2_), 2.41–2.48 (1H, m, CH), 2.86–2.92 (1H, m, CH), 3.09–3.18 (2H, m, CH_2_), 3.57–3.66 (1H, m, CH_2_), 3.85–3.94 (1H, m, CH_2_), 4.37−4.47 (1H, m, CH_2_), 4.51–4.59 (1H, m, CH_2_), 6.16−6.20 (1H_Py_, m), 6.20−6.25 (1H_Py_, m), 7.28−7.35 (1H_Py_, m); ^13^C NMR (DMSO-*d6*, 125 MHz) δ 26.13 (CH_2_), 27.68 (CH), 28.38 (1CH+3CH_adam_), 34.75 (CH), 36.49 (3CH_2adam_), 38.74 (3CH_2adam_), 48.85 (CH_2_), 50.39 (CH_2_), 51.94 (CH_2_), 105.48, 116.49 (1CH_Py_), 139.29 (1CH_Py_), 41.67, 150.10, 162.61, 175.95 (4C_quater_); MS m/z (I_rel_, %) 353.20 [M+H]^+^ (100), 375.20 [M+Na]^+^ (42.5), 705.40 [2M+H]^+^ (21.3), 727.40 [2M+Na]^+^ (58.0); Anal. calcd. for C_22_H_28_N_2_O_2_ (352.48): C, 74.97; H, 8.01; N, 7.95%; Found: C, 75.33; H, 8.15; N, 7.79%.

## 4. Conclusions

The convenient method for the preparation of the alkaloids anabasine and cytisine derivatives via an acylation reaction with 4,5-dichloroisothiazole-3-, 5-arylisoxazole-3-, adamantane- and hydrochlorides of pyridine-3- and pyridine-4-carbonyl chlorides, has been developed. Based on the synthesized derivatives, quaternary pyridinium salts (monoidomethylates and diiodomethylates) were obtained.

Analysis of the antimicrobial activity of the test substances showed that its manifestation depends on the type of pathogenic microorganism. The test strain *Staphylococcus aureus* is the most sensitive to almost all presented compounds, in particular to the derivatives of cytisine and anabasine with an isoxazole fragment. The obtained MIC data also allowed us to conclude that in some cases quaternization of *N*-acyl derivatives significantly increases antibacterial activity against *Staphylococcus aureus,* compared with the initial substrates.

The greatest analgesic effect, comparable to sodium diclofenac, was shown by the majority of anabasine derivatives, while cytisine derivatives at doses of 25 mg/kg did not show significant analgesic activity in the ″acetic writhing″ test.

According to influenza-virus testing for strains A/Almaty/8/98 (H3N2) and A/Vladivostok/2/09 (H1N1), only the anabasine and cytisine derivatives with adamantane fragment showed pronounced antiviral properties, exceeding even the activity of the commercial drugs Tamiflu and Remantadine.

The facts above makes it possible to consider *N*-acyl anabasine and cytisine derivatives promising for further study of their pharmacological properties. The leading candidates are planned to undergo in vivo testing and further modification of the structure, to achieve adequate clinical efficacy.

## Data Availability

The data presented in this study are available in the article or Appendix A.

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
