# Peer review of "Synthesis and Biological Activity of N-acyl Anabasine and Cytisine Derivatives with Adamantane, Pyridine and 1,2-Azole Fragments"

_molecules, 2022, doi:10.3390/molecules27217387_

Round 1

Reviewer 1 Report (Previous Reviewer 2)

The authors adequately addressed my comments in this final revised manuscript version. In this regard, the manuscript looks much better and improved in quality, content, and organization. 

Author Response

Dear reviewer! We sincerely appreciate all valuable comments and suggestions, which helped us to improve the quality of the manuscript.

Reviewer 2 Report (Previous Reviewer 1)

The authors have provided all necessary changes, so now the manuscript is ready for the acceptance.

Author Response

Dear reviewer! We sincerely appreciate all valuable comments and suggestions, which helped us to improve the quality of the manuscript.

This manuscript is a resubmission of an earlier submission. The following is a list of the peer review reports and author responses from that submission.

Round 1

Reviewer 1 Report

The article submitted for peer review is devoted to an important topic, the modification of alkaloids for the creation of drugs. Despite the importance of the research and their incomparable practical significance, the text of the article is written carelessly, poorly proofread, and sloppy (especially experimental part). I recommend this manuscript to be accepted after major revision.

1. The title should reflect that there were biological evaluations of the synthesized compounds.

2. Table 1 – replace all Russian symbols and add data on calculations (type of basis, type of calculations, program and etc.; a paragraph on quantum chemical calculations must be added to the experimental).

3. NMR spectra plots of the synthesized compounds must be added as SI.

4. Lines 245 and 247 – doi must be added as references.

5. In part about molecular docking, data on the binding or docking site must be added. Comments on cognate ligand are also required.

6. In part about molecular docking, text must be added about the interactions of the docked molecules with the target (what bonds are formed, are they good or bad for docking and etc., may be you will find any correlations with the bioexperimental data).

Author Response

Dear reviewer!

Thank you for the constructive comments. We took into account all your suggestions. We really think that the article has become better after making all corrections. Here you will find some explanations for each of your questions.

Point 1. The title should reflect that there were biological evaluations of the synthesized compounds.

Response 1. The title has been corrected.

Point 2. Table 1 – replace all Russian symbols and add data on calculations (type of basis, type of calculations, program and etc.; a paragraph on quantum chemical calculations must be added to the experimental).

Response 2. The paragraph to the experimental part was added.

Point 3. NMR spectra plots of the synthesized compounds must be added as SI.

Response 3. It's not clear. SI is the initial file format of the NMR viewer, isn¢t it?

Point 4. Lines 245 and 247 – doi must be added as references.

Response 4. Have done

Point 5. In part about molecular docking, data on the binding or docking site must be added. Comments on cognate ligand are also required.

Response 5. We inserted additional information to the experimental part

Point 6. In part about molecular docking, text must be added about the interactions of the docked molecules with the target (what bonds are formed, are they good or bad for docking and etc., may be you will find any correlations with the bioexperimental data).

Response 6. Unfortunately, at that moment we have received only this information from our colleague. Perhaps this section should be excluded from the article at all.

Reviewer 2 Report

The manuscript in reference describes the synthesis and biological activity evaluation of a set of fifteen derivatives of anabasine and cytisine. The manuscript has relevant information and results that will be interesting for readers. However, the presentation of results and discussion is not correctly structured, and the experimental details are not completely provided. Therefore, this manuscript has several issues that reduce its quality and needs to be addressed prior to further consideration.

Specific points:

1.       Detailed scrutiny should be performed throughout the manuscript to revise/correct several grammar and stylistic issues.

2.       Title: The title must be improved since it reflects the chemistry part of the manuscript, but the biological evaluation is not contemplated in the title.

3.       Abstract: A conclusive sentence should be added at the abstract ending.

4.       Lines 53-55. This passage appeared incomplete, so it can be improved by adding the proper complement. For instance, why is it a very urgent task? The scientific problem is unclear.

5.       Lines 56-58: This part is not adequately developed in the manuscript since the influence of the nature of the acyl fragment on biological activity is not clearly discussed and demonstrated in the Results & Discussion (R&D) section.

6.       Lines 70-71: This last sentence has no sense for the introduction section. It should be removed.

7.       The R&D section must be improved a lot since the information is highly descriptive, and the discussion is missing or even lacking. No comparisons with other reported studies and results are included and discussed. In addition, several paragraphs are short ideas without good writing development.

8.       Line 74: The enantiomeric form or optical activity (or even if the authors used a racemic mixture) of starting materials (i.e., anabasine and cytisine) must be provided and informed in the manuscript. Be consistent throughout the manuscript.

9.       Scheme 1: Revise this scheme since some mistakes are present. For instance, the positive charge of the quaternary pyridinium nitrogen is not totally visible, and bonds around them are not correctly sketched. In addition, the X substituents within the dotted box are not correctly aligned.

10.   Table 1. Revise the title of the fourth column since it has different language characters. In addition, the significant figures of the energies should be reduced. Finally, the energy units are missing and must be informed in this table.

11.   Lines 112-113: This passage is very laconic. The spectroscopic characteristics must be adequately described and developed. For instance, the global characteristic signals and information leading to the structural elucidation of the synthesized compounds must be provided.

12.   Line 129: How was defined the antibacterial effect of the compounds? MIC? Specify that. Be consistent throughout the manuscript.

13.   Table 2. The dose for these halo-based results must be provided. In addition, the decimal figures of mean values must be added to support the standard deviation values (±). A table caption is missing to self-explain the information listed in this table. For instance, the information of the number replicates is missing. Or, what does mean the hyphen (–)? Not determined, not evaluated, not active? Specify that. The identical issue for Table 3.

14.   Line 136: This information must be expanded for readers since it is laconically mentioned. A more profound development of this sentence should add a good discussion. Be consistent throughout the manuscript.

15.   Information between Figure 2 and Table 3 (first column) is repeated, which is inappropriate in a manuscript.

16.   Table 4: The decimal separators must be unified since some numbers have commas and others have dots. A similar issue is presented in Table 5 since the decimal separator is a comma. Be consistent throughout the manuscript.

17.   Line 176-177: This is another example of laconic passages (as mentioned in comment 11). Several short ideas are placed without connectivity, add-on, or precise aim. Several passages have the same issue and must be corrected. Otherwise, the manuscript will not be precise and will be challenging to follow.

18.   Line 178: A brief description of the steps and several experiments performed for the antiviral activity must be added. This part is not clear.

19.   Figure 3, 4: The axes (X and Y) legends are missing.

20.   Line 203: The number of this subheading is wrong.

21.   Line 243: The name of this subheading must be expanded to be self-explanatory. In addition, the numbering should belong to the previous section related to the antiviral activity (i.e., 2.2.3.6.).

22.   The discussion about molecular docking results is not appropriate. A deeper discussion should be provided.

23.   Line 280: A reference to support this final idea is missing.

24.   Revise in detail the M&M section. Some experimental details are missing to ensure outcome reproducibility. For instance, the brand, model, and grade of reagents, solvents, materials, and instruments must be provided.

25.   Line 305: The synthetic par must be located in this section and removed from the biological activity section.

26.   Line 308: This text has a different format (capital letters).

27.   Line 362: milliliters should be abbreviated as “mL” instead of “ml”. Be consistent throughout the manuscript.

28.   Line 445: The molecular docking procedure is missing. This procedure must inform several details such as ligand preparation, protein preparation, the active site coordinates, grid spacing and size, protocol validation, convergence evaluation, and control (co-crystallized native ligand or known inhibitor).

29.   The conclusion section should also be improved since it comprises a summary of the results. Authors should conclude the conceptual findings from a mechanistic point of view and even the scope of these results for future studies.

30. The information compiled as supplementary material was not provided in the additional documents. Furthermore, it is not clearly specified, listed and/or detailed in line 704.

Author Response

Dear reviewer!

Thank you for the constructive comments. As far as possible, we have taken into account all your suggestions. We really think that the article has become better after making all corrections. Here you will find some explanations for each of your questions.

Round 2

Reviewer 1 Report

SI is for supporting information.

The authors should provide NMR plots in SI (any format, e.g. doc, pdf and etc.).

Author Response

Dear reviewer,

Sorry for being inattentive about the Supplementary Materials. We have attached the NMR, IR, MS data.

Reviewer 2 Report

The authors addressed some of my comments. However, other major points were not adequately addressed and even were not replied. In addition, despite the results being interesting, the manuscript still has several shortcomings. Major concerns are again related to the manuscript structure, presentation of results, organization of ideas, and even the aim and scope. The molecular docking results are not properly discussed and, as mentioned by the authors, can be removed. The manuscript is highly descriptive, the findings are not properly informed, and several passages and ideas are unconnected or out of context. Indeed, some results are not properly described and discussed. Figures still having no captions for the X and Y axes. Synthesized compounds 2a-e and 3a-c,f should be optically active (based on the starting compound (-)-anabasine and (-)-cytisine), but optical rotations (at least) of circular dichroism measurements are not provided and discussed. Conclusions are laconically described and summarize results. No supplementary data (NMR, IR, MS, spectra) is provided for review. 

Author Response

Dear reviewer,

Taking into account your comment, we tried to correct the significant shortcomings of the article.

We are aware that the number of compounds we have obtained is not enough to establish the influence of the nature of the pharmacophore moiety on the biological activity of the target compound, so this idea was excluded from the introduction part. For two years our laboratory, in cooperation with colleagues from Kazakhstan has been searching for new biologically active compounds with a different spectrum of activity among alkaloid derivatives. Extending our previous studies on the synthesis of novel derivatives based on quinine alkaloid, we used the classical acylation reaction of anabasine and cytisine to obtain derivatives with 1,2-azole, pyridine and adamantane fragments, which are well-known pharmacophore fragments, especially since the methods for the synthesis of 1,2-azole derivatives, including those based on own author's methods, have been developing in our laboratory for many years. This time we have tried to explain more clearly our scientific background for this kind of research in the introduction part.

As for the comments on the presentation of the results, we believe that we have discussed the observed phenomena in detail. Could you specifically point out the inaccuracies in the discussion on your opinion? We were able to see some regularities, which are discussed in the final part. We cannot say for sure that the introduction of one or another fragment will definitely affect the appearance of a certain type of activity, since this requires a larger database of compounds, but we were able to choose the most promising agents from our synthesized.

As regards the optical activity of the target compounds, the device for measuring the angle of rotation was under repair for a long time. We found an opportunity and provided this data now.

We also excluded the part on molecular docking and the study of hemagglutination and neurominidase activity, since these data really need to be studied more carefully in order to present clear explanations.

Sorry for being inattentive about the Supplementary Materials. We have attached the NMR, IR, MS data.

We hope for your positive decision regarding our article.

Looking forward to your reply. Thank you in advance.

Ekaterina Akishina